# Phylogenetic Analysis to Explore the Association Between Anti-NMDA Receptor Encephalitis and Tumors Based on microRNA Biomarkers

**DOI:** 10.3390/biom9100572

**Published:** 2019-10-05

**Authors:** Hsiuying Wang

**Affiliations:** Institute of Statistics, National Chiao Tung University, Hsinchu 30010, Taiwan; wang@stat.nctu.edu.tw

**Keywords:** anti-NMDA receptor encephalitis, biomarker, microRNA, tumor, teratoma

## Abstract

MicroRNA (miRNA) is a small non-coding RNA that functions in the epigenetics control of gene expression, which can be used as a useful biomarker for diseases. Anti-NMDA receptor (anti-NMDAR) encephalitis is an acute autoimmune disorder. Some patients have been found to have tumors, specifically teratomas. This disease occurs more often in females than in males. Most of them have a significant recovery after tumor resection, which shows that the tumor may induce anti-NMDAR encephalitis. In this study, I review microRNA (miRNA) biomarkers that are associated with anti-NMDAR encephalitis and related tumors, respectively. To the best of my knowledge, there has not been any research in the literature investigating the relationship between anti-NMDAR encephalitis and tumors through their miRNA biomarkers. I adopt a phylogenetic analysis to plot the phylogenetic trees of their miRNA biomarkers. From the analyzed results, it may be concluded that (i) there is a relationship between these tumors and anti-NMDAR encephalitis, and (ii) this disease occurs more often in females than in males. This sheds light on this issue through miRNA intervention.

## 1. Introduction

Anti-NMDA receptor (anti-NMDAR) encephalitis, which was described and defined by Dalmau and colleagues [1], is an acute disease caused by the body’s own antibodies attacking N-methyl-D-aspartate (NMDA) receptors in the brain. Diagnosis is based on finding specific antibodies in the cerebral spinal fluid. Psychiatric symptoms and neurological disturbances including memory disturbances, seizures, dyskinesia, and catatonia develop during the progress of this disease. Due to initial psychiatric symptoms, it is not easy to accurately diagnose this disease in an early stage. Early treatment can lead to a good recovery outcome. The disease is more prevalent in women, and about 37% of patients are younger than 18 years [2]. The cause of this disease is usually unknown. Tumors, especially ovarian teratomas, have been detected in a proportion of patients [3,4]. Patients with detectable tumors had significant improvement after tumor resection. The cause of this disease is often unknown for most patients without detectable tumors. Vaccination may induce this disorder [5,6,7]. In addition, patients with herpes simplex encephalitis might produce antibodies against NMDA receptors [8,9], and this disease might be induced by other viruses [10]. More updated discussion and review of this disease including animal models are provided by the experts of this field [2].

Full recovery of this disease can take from several months to several years after disease onset. Most patients with anti-NMDAR encephalitis respond to immunotherapy and the immunotherapies used, timing of improvement, and long-term outcome have been studied [11]. The first-line of immunotherapies for anti-NMDAR encephalitis include steroids, intravenous immunoglobulin (IVIG), and plasma exchange (or plasmapheresis); the second-line of immunotherapies include rituximab and cyclophosphamide. A treatment strategy using at least two of these therapies may lead to higher efficacy rates than treatment with only a single form of therapy [12]. Treatment efficacy may differ by gender [13]. Patients with tumors are treated with tumor resection. Therefore, the treatment strategy may be a potential factor in facilitating an early recovery from anti-NMDAR encephalitis. In addition, a grading score predicting neurologic function one year after diagnosis of anti-NMDAR encephalitis was constructed [14]. The functional status of 382 patients one year after diagnosis was studied, and the factors associated with poor status were identified.

The underlying mechanism of the anti-NMDAR encephalitis is that the GluN1 subunit of the NMDA receptors in the brain is targeted by autoantibodies [15]. This may be induced by cross-reactivity with NMDA receptors in teratomas that contain brain cells. This finding suggests that tumors may trigger the anti-NMDAR immune response [4]. The removal of an ovarian cystadenofibroma may lead to a full resolution of this disease. The serum NMDAR antibodies of an adolescent female patient disappeared after the removal of an ovarian cystadenofibroma [16]. A prompt neurological response in a patient resulted from early removal of an ovarian teratoma followed by plasma exchange and corticosteroids [17]. A patient had good recovery of consciousness after tumor removal [18]. In a long-term follow-up in the absence of tumor resection in four Japanese women, the severity and extended duration of symptoms supported tumor removal [19]. From these clinical data analyses, strong evidence showed that good recovery was attained after tumor resection. More studies associating anti-NMDAR encephalitis with ovarian teratomas are listed in Table 1.

In addition to ovarian teratomas, many cases show that other tumors may trigger the anti-NMDAR immune response. Hepatic neuroendocrine carcinoma is associated with this disorder [20]; anti-NMDAR encephalitis is associated with a large-cell neuroendocrine carcinoma [21]; anti-NMDAR encephalitis may be related to matter lesions [22,23]; an 18 year old female case was caused by a large mature mediastinal teratoma [24]; a right anterior mediastinal mass with no adenopathy was found in a female patient [25]; an anti-NMDAR encephalitis male patient had a testicular teratoma and seminoma [26]; a case report showed the association between anti-NMDAR encephalitis and small cell lung carcinoma [27]. More references to the associations between this disease and tumors are listed in Table 1.

Anti-NMDAR encephalitis can also be induced by virus infection. Herpes simplex type 1 encephalitis (HSE) was reported to induce anti-NMDAR encephalitis in many cases. A portion of children, after recovery from HSE encephalitis, developed anti-NMDAR encephalitis [8,28,29]. NMDA receptor antibodies were identified in patients with relapsing post HSE [28]. Patients having a relapsing course after HSE had elevated autoantibodies against the NMDA receptor [8].

In this paper, I focus on exploring the association between anti-NMDAR encephalitis and tumors. To the best of my knowledge, there have not been any studies in the literature investigating this issue through their microRNA (miRNA) biomarkers. I adopt a phylogenetic analysis to plot the phylogenetic trees of their miRNA biomarkers. From the analyzed results, I conclude that there is a relationship between anti-NMDAR encephalitis and a number of tumors. In addition, these phylogenetic trees can explain the fact that this disease occurs more often in females than in males. This sheds light on this issue through miRNA intervention.

This study was approved by the Research Ethics Committee for Human Subjects at National Chiao Tung University, Taiwan. In this paper, I first review miRNA biomarkers for anti-NMDAR encephalitis and tumors, and then I discuss the tumors related to anti-NMDAR encephalitis, which was obtained by searching the literature. Finally, I adopt a phylogenetic analysis to analyze these miRNA biomarkers. The phylogenetic analysis has been used to study miRNA functions in the literature. A study showed that microarray analysis combined with the phylogenetic tree analysis can improve the accuracy of miRNA biomarker prediction compared with a method based only on microarray analysis [30]; the evolution of the two disease resistance-related miRNAs was inferred using the phylogenetic analyses [31], and older miRNAs were significantly more likely to be associated with disease than younger miRNAs from the phylogenetic analysis [32]. In addition, the phylogenetic tree has been used as a tool to analyze the relationship between vaccination and anti-NMDAR encephalitis based on their miRNA biomarkers [5]. Thus, I apply phylogenetic analysis in this study.

In addition to miRNA biomarkers, other responsible factors for autoimmune diseases may be used to explore the association between anti-NMDAR encephalitis and tumors in a future study. For this disease, miRNA biomarkers have been used to investigate the association between anti-NMDAR encephalitis and vaccination based on the phylogenetic tree analysis. The phylogenetic tree method can increase the accuracy of tumor miRNA biomarker prediction. In addition, miRNA has been shown to play a key role in tumor mechanisms. Thus, I focus here on miRNA biomarker research.

## 2. MicroRNA

Abnormal epigenetic changes have been demonstrated to be associated with human diseases [33,34]. miRNA is a small non-coding RNA that functions in epigenetic control of gene expression [35], and epigenetic mechanisms can also affect the expressions of miRNAs [36]. miRNAs have been shown to be linked to cancer, and they play a role as either tumor-suppressor genes or oncogenes in carcinogenesis. miRNA biomarkers for cancer have been explored [30,37,38]. In addition to cancer, miRNAs are also associated with many other diseases such as neurodegenerative diseases, hematological diseases, and autoimmune diseases [39,40,41,42,43,44].

### 2.1. Anti-NMDA Receptor Encephalitis and miRNAs

There are only a few studies investigating miRNA expression for anti-NMDAR encephalitis. The miRNA let-7 family was demonstrated to be associated with anti-NMDAR encephalitis. Let-7 was one of the first miRNAs discovered in the nematode, which is complementary to elements in the 3’ untranslated regions of genes lin-14, lin-28, lin-41, lin-42, and daf-12. This indicates that let-7 may directly control these genes [44].

Loss of function for let-7 has been observed in many human cancers, including lung, liver, stomach, esophageal, colon, prostate, breast, and ovarian cancers [45,46,47]. Let-7 plays a role as a tumor suppressor in cancer cells by targeting genes [48]. Let-7a, let-7b, let-7c, and let-7i have significantly different expression levels in atretic porcine ovary follicles compared with healthy follicles [49], and let-7e is associated with autoimmune encephalomyelitis [50].

Members of the let-7 family, including let-7a, let-7b, let-7d, and let-7f, were found to be down-regulated in anti-NMDAR encephalitis compared with the controls [51]. The anti-NMDAR encephalitis patients in that study were recruited at the prodromal phase and psychotic phase before immunotherapy. Fresh blood samples were acquired from these patients and controls. The miRNAs expression levels of five anti-NMDAR encephalitis plasma and five control plasma were quantitatively tested using microarray analysis. Blood samples that were used for real-time quantitative polymerase chain reaction (qRT-PCR) were obtained from anti-NMDAR encephalitis patients and controls (Figure 1). In addition, blood samples from patients for other nervous system diseases were also compared using qRT-PCR analysis. As a result, the expression levels of the three miRNAs, let-7a, let-7d, and let-7f, in other nervous system diseases were significantly down-regulated compared with the normal control group. Only let-7b did not have a significantly different expression level in other nervous system diseases compared with the normal control group [51]. Thus, they concluded that microRNA let-7b has the potential to be used as a biomarker for anti-NMDAR encephalitis.

The result of let-7 family association with the anti-NMDAR encephalitis has been used as a tool for analyzing the relationship between this disease and vaccination. Wang [5] reported an anti-NMDAR encephalitis case related to Japanese encephalitis (JE) vaccination and plotted phylogenetic trees of miRNAs relating to anti-NMDAR encephalitis or vaccines by using the let-7 family as a biomarker of anti-NMDAR encephalitis [5,52]. In addition to miRNA let-7b, the B-cell attracting C–X–C motif chemokine 13 was discussed as a biomarker for anti-NMDAR encephalitis [5,53,54].

Another study used an in vitro model for exploring miRNA biomarkers. The dysregulation of miRNA and mRNA expression in the in vitro anti-NMDAR encephalitis model was investigated [55]. Researchers incubated hippocampal neurons of rat pups with cerebrospinal fluid from four anti-NMDAR encephalitis patients and four negative controls. Four miRNAs, including miR-139-3p, miR-6216, miR-135a-3p, and miR-465-5, were identified to be differentially expressed in anti-NMDAR encephalitis in the in vitro model.

### 2.2. Tumors Associated with Anti-NMDAR Encephalitis and miRNAs

A literature search was conducted for studies reporting tumors associated with anti-NMDAR encephalitis. The references of these studies and miRNAs associated with these tumors are presented in Table 1.

The literature review shows that many female anti-NMDAR encephalitis patients had ovarian teratomas. The association between ovarian teratomas and miRNAs was discussed; miR-421, miR-555, miR-492, and miR-26b* were highly expressed in mature ovarian teratoma tissues, whereas let-7a, miR-19a, miR-34a, miR-620, miR-214, miR-142-3p, miR-934, miR-720, miR-22, miR-629, and miR-657 were less expressed compared with normal tissues [70]. In addition to ovarian teratomas, the let-7 family is related to ovarian cancer. For ovarian cancer, the copy number of let-7b and mature let-7b expression had a positive correlation, and let-7b expression was shown to be related to ovarian tumor growth both in vitro and in vivo [90]. Let-7 was associated with poor prognosis of ovarian cancer [91]. Low levels of let-7a-3 expression were associated with a poor prognosis in epithelial ovarian cancer [92].

Neuroendocrine tumors (NETs), which are neoplasms developed from neuroendocrine cells and nervous systems, were found in anti-NMDAR encephalitis patients. miR-129-5p and the let-7 family were down-regulated in NETs or neuroendocrine tumor metastases [75]; miR-29b-3p, miR-21-5p, miR-150-5p, and miR-22-3p had different expression levels in NET cases and controls [76]. In pancreatic NETs, an upregulation of miR-103 and miR-107 was identified [77]. miR-196a was detected in NETs and poor prognosis in resected pancreatic NETs was associated with a high expression of miR-196a [78,79].

Testis teratomas are associated with anti-NMDAR encephalitis for male patients. In addition, the let-7 family plays a role in the pathogenesis of testicular cancer [85]. Let-7a, let-7d, and miR-294 have different expression levels in germ-line stem cells compared with multipotent adult germ-line stem cells [84]. Estrogen receptors (ERα) have an essential role in male fertility, and mir-100 and let-7b may target the ERα gene [93].

Several anti-NMDAR encephalitis cases were related to small-cell lung cancer. The expression of let-7 was down-regulated in the human small-cell lung cancer (SCLC) cell line [88]. In addition, let-7 is also closely related to non-small cell lung cancer (NSCLC). A study of 143 lung cancer cases showed that patients with reduced let-7 expression had shorter survival [94]. The over-expression of cofilin-1 suppressed the growth and invasion of NSCLC cells in vitro. Let-7 is involved in over-expression of cofilin-1 that can suppress the growth of NSCLC cells in vitro and in vivo [95].

## 3. Materials and Methods

### 3.1. miRNA Sequence

The first part of the analysis was to find miRNA biomarkers for anti-NMDAR encephalitis and the related tumors, including ovarian teratoma, dura mater lesions, neuroendocrine tumor, mediastinal teratoma, testis teratoma, and small-cell lung cancer. I searched the literature using the keywords miRNA, anti-NMDAR encephalitis, teratoma, and tumor to find the related studies or case reports. The miRNA biomarkers of these diseases are listed in Table 1. To apply the phylogenetic analysis method, I first obtained the nucleotide sequences of these miRNAs from miRBase [96]. miRBase is a database of microRNA sequences and annotations. The stem-loop sequences of miRNAs accessed from miRBase were used to plot the trees because the stem-loop sequence can provide more information than the mature -5p sequence and mature -3p sequence. As seen in Table 1, the sequence of miR-294, one of the testis teratoma miRNA biomarkers, could not be found in miRBase, and thus, I did not include this miRNA in the phylogenetic trees. In addition, since the let-7 family members are involved in most of the tumors shown in Table 1, to find the relationship between anti-NMDAR encephalitis and these tumors, I discussed let-7 additionally and did not include let-7 as a biomarker of these tumors in the plotted phylogenetic trees.

### 3.2. Phylogenetic Analysis

A phylogenetic analysis was adopted to investigate the association between these tumors and anti-NMDAR encephalitis using the miRNA biomarkers listed in Table 1. The phylogenetic tree is one of the most useful phylogenetic analysis tools that can be used to explore the evolutionary relationship between nucleotide sequences. The phylogenetic tree of nucleotide sequences shows the relationship among various nucleotide sequences based upon similarities and differences of their genetic characteristics. The similarities of two nucleotide sequences can be measured using different substitution models [97,98].

To plot the phylogenetic tree of miRNAs, I used the bioinformatics toolbox of the MATLAB software [99]. Using this software, I first needed to select a substitution model (distance method) to calculate the pairwise distances between any pair of these nucleotide sequences and then choose a clustering method to plot the phylogenetic trees. The substitution model for calculating the sequence distance in the MATLAB software includes the p-distance, Jukes–Cantor distance, and alignment-score distance; the clustering method in the MATLAB software used to build the tree includes the median method, the single method, and the average method. The MATLAB code for performing the first step is “seqpdist,” and the code for performing the second step is “seqlinkage.” An example code of plotting a phylogenetic tree of three miRNAs is provided in the Appendix A.

In this paper, I used the Jukes–Cantor distance (or the alignment-score distance, p distance) to calculate the distances and the median method (or the average method) as the clustering method to build the trees.

## 4. Results

Table 1 shows the references that reported anti-NMDAR encephalitis related to ovarian teratoma, dura mater lesions, neuroendocrine tumor, mediastinal teratoma, testis teratoma, and small-cell lung cancer. It also shows the references for the miRNA biomarkers of these tumors except for the dura mater lesions and mediastinal teratoma. I did not find any miRNA biomarkers for dura mater lesions and mediastinal teratoma. As seen in Table 1, the let-7 family involves four types of tumors. As mentioned in the Introduction, let-7a, let-7b, let-7d, and let-7f can be potential biomarkers of anti-NMDAR encephalitis. Thus, based on the fact that this disease and the four types of tumors have the common let-7 biomarker, it may be concluded that this disease is related to these four types of tumors from their miRNA biomarkers. This result coincides with the reported cases that anti-NMDAR encephalitis is associated with these tumors.

To explore more mechanisms linking this disease to tumors, the phylogenetic trees were plotted using the miRNA biomarkers listed in Table 1. Figure 2a presents the phylogenetic trees that were plotted using the Jukes–Cantor model distance and the average method; Figure 2b presents the phylogenetic trees that were plotted using the Jukes–Cantor model distance and the median method. Figure 3a presents the phylogenetic trees that were plotted using the alignment-score distance and the average method; Figure 3b presents the phylogenetic trees that were plotted using the alignment-score distance and the median method. Figure 4a presents the phylogenetic trees that were plotted using the p distance and the average method; Figure 4b presents the phylogenetic trees that were plotted using the p distance and the median method.

As seen in Figure 2, Figure 3 and Figure 4, the three testis teratoma biomarkers, miR-371, miR-372, miR-373, were not as close to the let-7 family as other tumors. Especially, as seen in Figure 3, these three miRNAs and let-7 were significantly separated into two branches. Since I adopted different evolutionary models and clustering methods to produce these figures, all of them show similar results. Based on the finding that testis teratoma biomarkers are not as close to let-7 as ovarian teratoma biomarkers, I may conclude that there is a gender difference in the prevalence of this disease. This coincides with the fact that anti-NMDAR encephalitis occurs more often in females than in males.

For the other three types of tumors, namely ovarian teratomas, neuroendocrine tumor, and small-cell lung cancer, as seen in Figure 2, Figure 3 and Figure 4, a number of their miRNA biomarkers are close to the let-7 family for each of them. This reveals that these three types of tumors are related to anti-NMDAR encephalitis through their miRNA biomarker phylogenetic analysis. In addition, although the three testis teratoma biomarkers are not as close to the let-7 family as the other types of tumors, they are not always in the farthest branch of the let-7 family in these figures. Therefore, from this phylogenetic analysis, these miRNA biomarkers cannot be excluded as being related to the let-7 family.

## 5. Discussion

There are many case reports revealing that anti-NMDAR encephalitis is associated with tumors, especially ovarian teratomas. In this study, I investigated their relationship using the phylogenetic trees of their miRNA biomarkers. As mentioned in the Results section, the ovarian teratoma, neuroendocrine tumor, testis teratoma, and small-cell lung cancer have let-7 family members as their biomarkers, as well as the anti-NMDAR encephalitis. Let-7 is involved in at least 15% of all human cancers, including lung, breast, liver, esophageal, stomach, ovarian, prostate, and colon cancers, as well as neuroblastoma and chronic lymphocytic leukemia [45,46,47,48]. As a result, if we use the let-7 family as biomarkers of anti-NMDAR encephalitis to investigate the relationship between anti-NMDAR encephalitis and tumors, then it is not surprising that this disease is highly related to many tumors. In addition, identifying more prognostic biomarkers and developing efficient treatments are the two most important issues in future studies [2].

Several members of the let-7 family, including let-7a, let-7b, let-7d, and let-7f, are related to anti-NMDAR encephalitis. However, some of these miRNAs are also associated with ovarian teratomas, neuroendocrine tumor, testis teratomas, and small-cell lung cancer, as seen in Table 1. Therefore, they could also be biological biomarkers of these tumors, since among these four miRNAs, only let-7b has no significantly different expression between other nervous system diseases and normal control. Thus, let-7b could be a potential biomarker of anti-NMDAR encephalitis [51]. However, more studies comparing the expression level of let-7b in patients with tumor and in patients without tumor are essential to assert that the expression level of let-7b can be identified as a biomarker of anti-NMDAR encephalitis but is not affected by the tumors of patients.

In addition to discussing the gender difference in the prevalence of anti-NMDAR encephalitis from the phylogenetic tree analysis, a number of studies discussed gender difference in the treatment strategy of this disease. The data of 94 patients were analyzed and the result showed that female patients had a higher efficacy rate than male patients [12]. In the comparison of the two treatments, IVIG and plasma exchange (or plasmapheresis), the female patients receiving IVIG had a higher efficacy rate than male patients. On the contrary, the efficacy rate of the plasmapheresis (or plasma exchange) was not inferior to IVIG for male patients [13]. The results of these studies may suggest the need to explore the mechanism of this disease’s underlying gender difference.

## 6. Conclusions

Anti-NMDAR encephalitis is an autoimmune disease. Although there are several types of immunotherapies, including steroids, intravenous immunoglobulin, plasma exchange (or plasmapheresis), rituximab, and cyclophosphamide, tumor resection is a more effective treatment than other immunotherapies when a tumor is detected in a patient [94]. This reveals that there is a close relationship between the tumor and anti-NMDAR encephalitis. The let-7 family is a miRNA biomarker of the tumors related to anti-NMDAR encephalitis. In addition, the expression of let-7 is also altered in anti-NMDAR encephalitis patients compared with healthy individuals. As a result, it may be unclear whether the altered expression level of the let-7 family is caused by the tumor or anti-NMDAR encephalitis. Thus, although let-7b could be a potential serum biomarker for anti-NMDAR encephalitis, more experiments comparing both anti-NMDAR encephalitis patients with tumor and patients without tumor are needed to clarify whether the altered expression level of let-7b is due to this disease or the tumor associated with this disease.

In addition, from these phylogenetic trees, the testis teratoma biomarkers are not as close to the anti-NMDAR encephalitis biomarker as are other tumors. Thus, from these results, by comparing the testis teratoma biomarkers and ovarian teratoma biomarkers, it may be concluded that ovarian teratomas have a higher chance of inducing anti-NMDAR encephalitis than testis teratomas. This coincides with the fact that anti-NMDAR encephalitis occurs more often in females than in males.

## Figures and Tables

**Figure 1 biomolecules-09-00572-f001:**
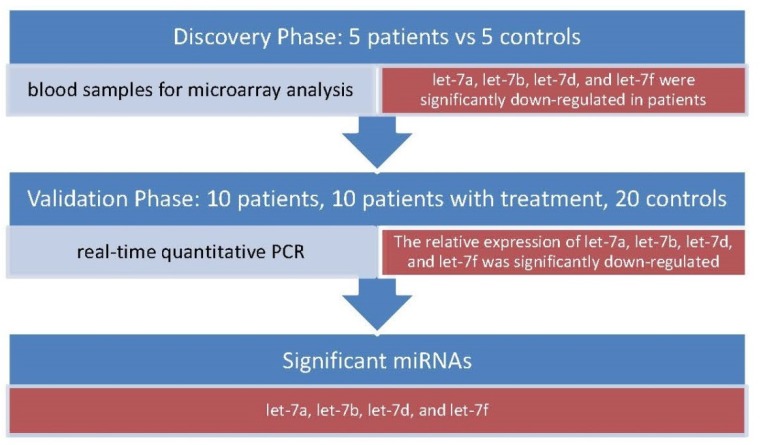
Let-7a, let-7b, let-7d, and let-7f were discovered to be miRNA biomarkers of anti-NMDAR encephalitis.

**Figure 2 biomolecules-09-00572-f002:**
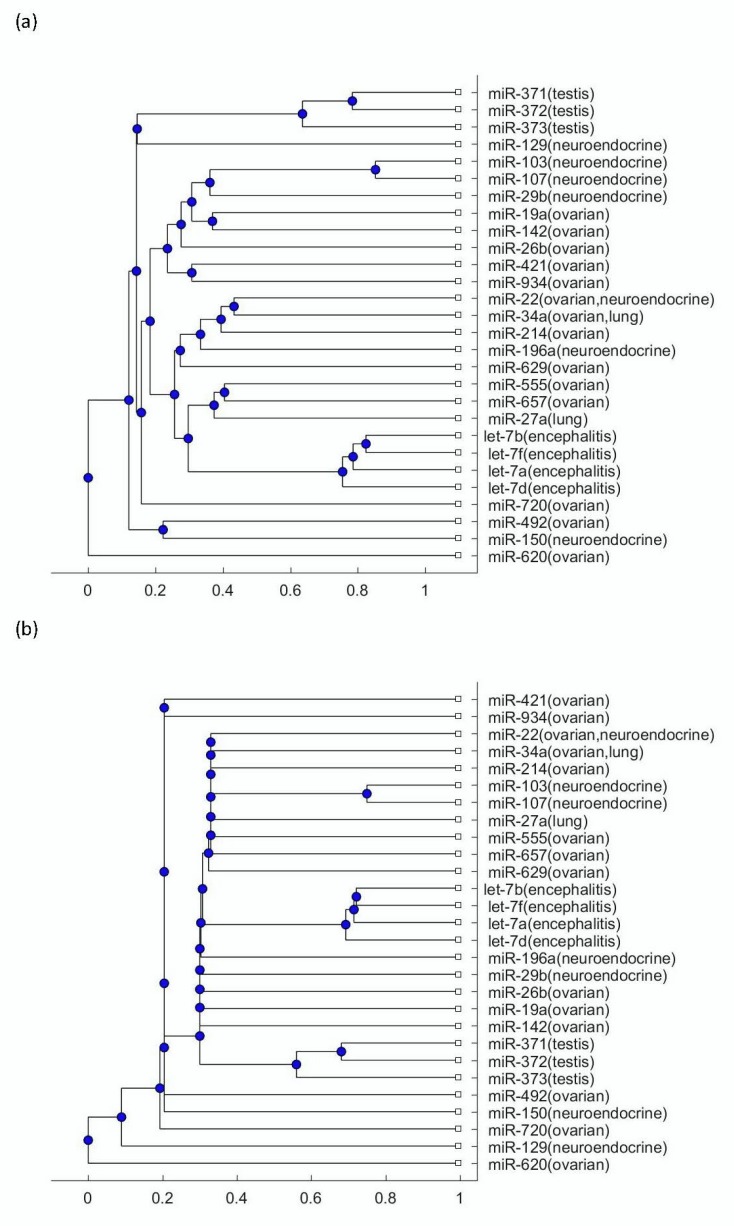
(**a**) The phylogenetic tree of miRNAs shown in Table 1 based on the Jukes–Cantor model distance and the average method. (**b**) The phylogenetic tree of miRNAs shown in Table 1 based on the Jukes–Cantor model distance and the median method.

**Figure 3 biomolecules-09-00572-f003:**
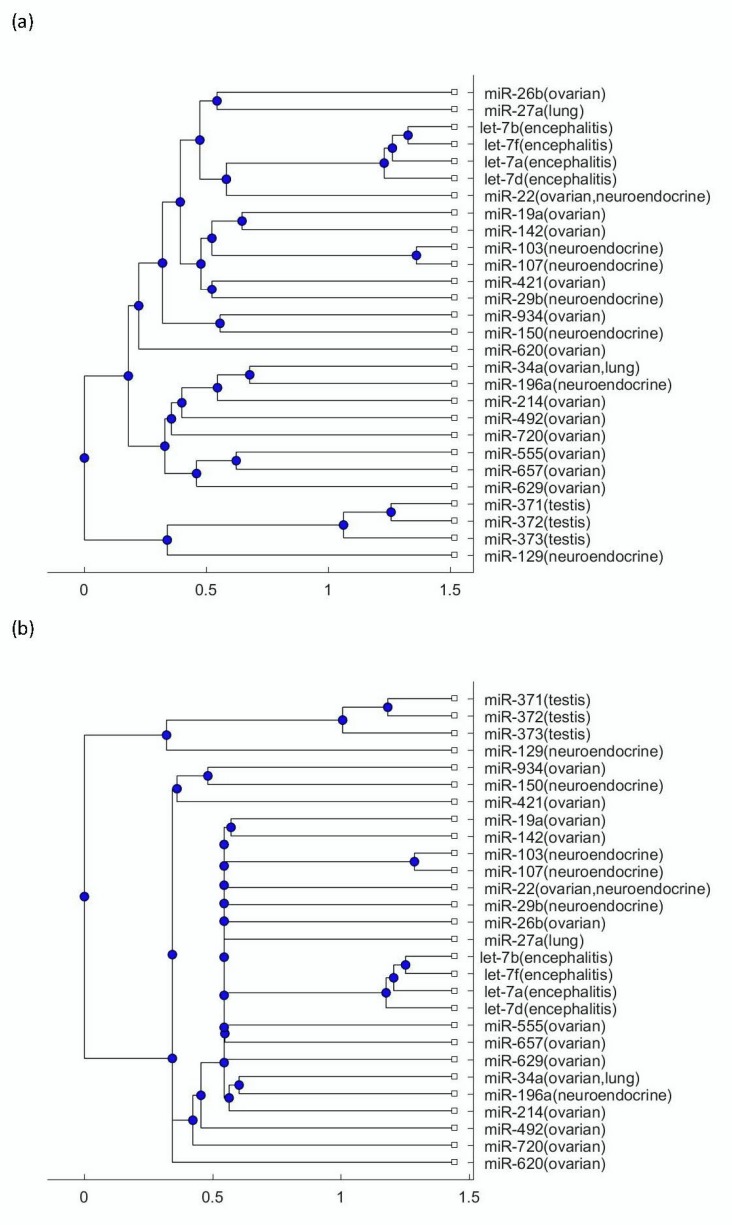
(**a**) The phylogenetic tree of miRNAs shown in Table 1 based on the alignment-score distance and the average method. (**b**) The phylogenetic tree of miRNAs shown in Table 1 based on the alignment-score distance and the median method.

**Figure 4 biomolecules-09-00572-f004:**
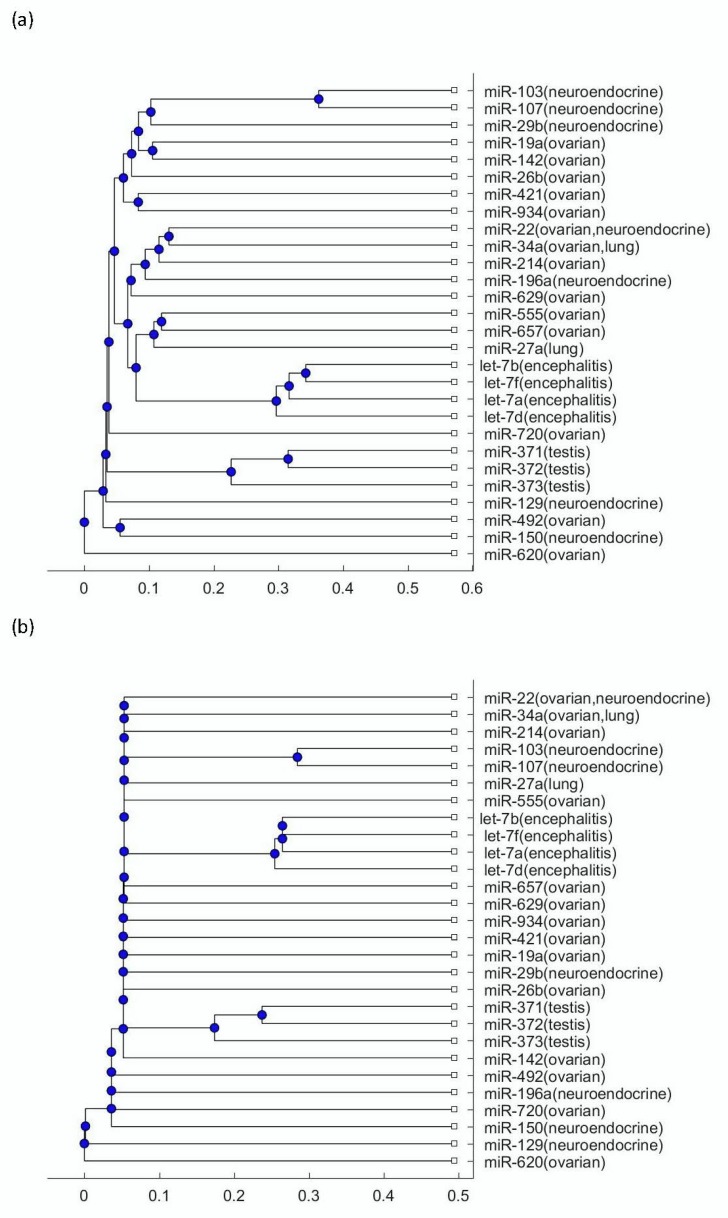
(**a**) The phylogenetic tree of miRNAs in shown Table 1 based on the p distance and the average method. (**b**) The phylogenetic tree of miRNAs shown in Table 1 based on the p distance and the median method.

**Table 1 biomolecules-09-00572-t001:** Tumors associated with anti-NMDAR encephalitis.

Tumor	Anti-NMDAR Encephalitis References	miRNAs	miRNA References
Ovarian teratoma	[4,16,17,18,56,57,58,59,60,61,62,63,64,65,66,67,68,69]	miRNA-26b*, miRNA-421, miRNA-22, miRNA-492, miRNA-555, miRNA-19a, miRNA-34a, miRNA-620, miRNA-142-3p, let-7a, miRNA-934, miRNA-657, miRNA-720, miRNA-629, miRNA-214	[70]
Dura mater lesions	[22,23,71]	-	-
Neuroendocrine tumor	[15,20,21,72,73,74]	miR-129-5p, let-7, miR-150-5p, miR-29b-3p, miR-22-3p, miR-21-5p, miR-103, miR-107, miRNA-196a	[75,76,77,78,79,80]
*Mediastinal teratoma	[24,25,81,82]	-	-
Testis teratoma	[26,83]	let-7a, let-7d, miR-294, miR-371, miR-372, miR-373	[84,85,86]
Small-cell lung cancer	[27,74,83,87]	let-7, miR-27a-5p, miR-34-5p,	[88,89]

*Mediastinal teratoma from chest scan.

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
