# Peer review of "Phylogenetic Analysis to Explore the Association Between Anti-NMDA Receptor Encephalitis and Tumors Based on microRNA Biomarkers"

_biomolecules, 2019, doi:10.3390/biom9100572_

Round 1

Reviewer 1 Report

This is an interesting paper, however the papers needs to improve interms of clarity and its identity, to spell ourt wheter it is a critical appraisal of the literature or secondary data analyses?

-The author must state in the text in detail whether he has obtained full Ethics Review Board approval for his work inclusing as applied, Preclinical Research Ethics Review approval.

-Methods section can be improved using subtitles.

-The title of the paper may need improvement and a descriptive title would be helpful.

Author Response

Thank you very much for the comments. I have revised the paper according to your comments. The point-by-point responses are provided as follows. All changes of this manuscript are highlighted with track changes in Word.

This is an interesting paper, however the papers needs to improve interms of clarity and its identity, to spell ourt wheter it is a critical appraisal of the literature or secondary data analyses?

Response: Thank you for your suggestion of changing the title of this manuscript.

The title was changed to ‘Phylogenetic analysis to explore the association between anti-NMDA receptor encephalitis and tumor based on microRNA biomarkers’. This title indicates that this manuscript focuses on data analysis. In Sections 2, I searched literature to find miRNA biomarkers and then used the method provided in Section 3 to analyze these data. The former title might mislead the type of this manuscript. I hope that the new title can clarify it.

-The author must state in the text in detail whether he has obtained full Ethics Review Board approval for his work inclusing as applied, Preclinical Research Ethics Review approval.

Response: This study was approved by the Research Ethics Committee for Human Subject at National Chiao Tung University, Taiwan (page 4, lines 6 and 7 ↑).

-Methods section can be improved using subtitles.

Response: The method section was moved to the third section. I used the two subtitles, miRNA sequence and phylogenetic analysis, in this section. Moreover, to provide more information about the MATLAB code, I added an example code in a supplementary file to show how to plot the phylogenetic tree using the MATLAB software.

-The title of the paper may need improvement and a descriptive title would be helpful.

Response: the title was changed to “Phylogenetic analysis to explore the association between anti-NMDA receptor encephalitis and tumor based on microRNA biomarkers”.

Reviewer 2 Report

In this manuscript, Dr. Wang clustered several miRNAs by using phylogeneitc tress and was trying to address the relationship between anti-NMDA receptor encephalits, tumor, and miRNA in the study. 

Major comments:

1.Please double check the author contributions and add authors to the author list if needed since Dr. Wang uses 'we' in the manuscript.

2. It will be of great help if the author can provide more detailed introduction/background why he focused on miRNAs as a biomaker, not the autoimmune responsible factors such as cytokines.

3. Although phylogenetic trees is widely used, it is one of the model. More data, ideally experimental data or strong evidence should be required to make a solid conclusion as claimed.

4. Related No.3, please provide figures why let-7 or related  miRNAs can be a biomaker in patients, by confirming wet experiment, related, or other dry analysis are essential.

Minor comments:

1. Please discuss more about the gender difference if you would like to focus on, could be regulated by sex hormones? If not, suggest to rewrite or omit. 

2. It could be Communicatoins rather than Article.

Author Response

Thank you very much for the comments. I have revised the paper according to your comments. The point-by-point responses are provided as follows. All changes of this manuscript are highlighted with track changes in Word.

Major comments:

1.Please double check the author contributions and add authors to the author list if needed since Dr. Wang uses 'we' in the manuscript.

Response: Since I am the sole author of this manuscript, I changed ‘we’ to ‘I’ in most cases. In some sentences, I used ‘we’ to denote readers and me.

It will be of great help if the author can provide more detailed introduction/background why he focused on miRNAs as a biomaker, not the autoimmune responsible factors such as cytokines.

Response: I added a paragraph to explain why I focus on miRNA biomarkers in this study.

Page 5, the second paragraph

‘In addition to miRNA biomarkers, other autoimmune responsible factors may be used to explore the association between anti-NMDAR encephalitis and tumors in a future study. For this disease, miRNA biomarkers have been used to investigate the association between anti-NMDAR encephalitis and vaccination based on the phylogenetic tree analysis. The phylogenetic tree method can increase the accuracy of tumor miRNA biomarker perdition. In addition, miRNA has been shown to play a key role in tumor mechanism. Thus, here I focus on miRNA biomarkers study. ‘

Although phylogenetic trees is widely used, it is one of the model. More data, ideally experimental data or strong evidence should be required to make a solid conclusion as claimed.

Response: The relationship between tumor and anti-NMDAR encephalitis has already been confirmed because many patients have significant recovery by resecting tumors. In this study, I apply the phylogenetic tree analysis to check whether we can use miRNA biomarkers to explain the association between tumor and anti-NMDAR encephalitis. I added more evidence from the literature to show this relationship.

Page 3, the last paragraph

‘This finding suggests that tumors may trigger the anti-NMDAR immune response [3]. The removal of an ovarian cystadenofibroma may lead to a full resolution of this disease. The serum NMDAR antibodies of an adolescent female patient were disappeared after the removal of an ovarian cystadenofibroma [14]. A prompt neurological response was resulted by early removal of an ovarian teratoma followed by plasma exchange and corticosteroids [15]. A patient had very good recovery of consciousness after tumor removal [16]. In a long-term follow-up in the absence of tumor resection in four Japanese women, the severity and extended duration of symptoms support tumor removal [17]. From these clinical data analyses, strong evidence showed that good recovery was attained after tumor resection.’

Related No.3, please provide figures why let-7 or related miRNAs can be a biomaker in patients, by confirming wet experiment, related, or other dry analysis are essential.

Response: I add a figure (Figure 1) of the experiment conducted in the reference paper [55]. This figure explains the experimental procedure that let-7a, let-7b, let-7d, and let-7f were discovered to be miRNA biomarkers of anti-NMDAR encephalitis. In addition, the way of the collection of blood samples was added in this version.

Page 6, the second paragraph

‘The anti-NMDAR encephalitis patients in that study were collected at the prodromal phase and psychotic phase before immunotherapy. Fresh blood samples were acquired from these patients and controls. The miRNAs expression levels of 5 anti-NMDAR encephalitis plasma and 5 control plasma were quantitatively tested using microarray analysis. Blood samples that were used for real-time quantitative polymerase chain reaction (qRT-PCR) were obtained from anti-NMDAR encephalitis patients and controls (Figure 1). In addition, blood samples from patients for other nervous system diseases were also compared using qRT-PCR analysis.’

Minor comments:

Please discuss more about the gender difference if you would like to focus on, could be regulated by sex hormones? If not, suggest to rewrite or omit.

Response: A paragraph was added to discuss more about the gender difference of this disease.

Page 16, the second paragraph

‘In addition to discussing the gender difference in the prevalence of anti-NMDAR encephalitis from the phylogenetic tree analysis, a number of studies had discussed the gender difference in the treatment strategy of this disease. The data of 94 patients were analyzed and the result showed that the female patients have a higher efficacy rate than the male patients [11]. In the comparison of the two treatments, IVIG and plasma exchange (or plasmapheresis), the female patients receiving IVIG had a higher efficacy rate than male patients. On the contrary, the efficacy rate of the plasmapheresis (or plasma exchange) is not inferior to IVIG for male patients [12].  The results of these studies may suggest exploring the mechanism of this disease underlying gender difference.’

It could be Communicatoins rather than Article.

Response: I do not have any opinion if the editor suggests that the paper is more suitable to be a communication paper than an article paper.

Reviewer 3 Report

This manuscript has not described the criteria of cases collection.

The information of the most important tool in this study “MATLAB” has not been described in this manuscript.

The style of this manuscript is not according to the instruction.

Author Response

Thank you very much for the comments. I have revised the paper according to your comments. The point-by-point responses are provided as follows. All changes of this manuscript are highlighted with track changes in Word.

This manuscript has not described the criteria of cases collection.

Response: The criteria of cases collection for discovering the miRNA biomarkers of anti-NMDAR encephalitis were provided in the reference paper [55]. I added some information and a figure (Figure 1) in this revised version.

Page 6, the second paragraph

 ‘The anti-NMDAR encephalitis patients in that study were collected at the prodromal phase and psychotic phase before immunotherapy. Fresh blood samples were acquired from these patients and controls. The miRNAs expression levels of 5 anti-NMDAR encephalitis plasma and 5 control plasma were quantitatively tested using microarray analysis. Blood samples that were used for real-time quantitative polymerase chain reaction (qRT-PCR) were obtained from anti-NMDAR encephalitis patients and controls (Figure 1). In addition, blood samples from patients for other nervous system diseases were also compared using qRT-PCR analysis.’

The information of the most important tool in this study “MATLAB” has not been described in this manuscript.

Response: I added an example MATLAB code in a supplementary file to show how to plot the phylogenetic trees using the MATLAB software.

The style of this manuscript is not according to the instruction.

Response: I reorganized the manuscript by moving the Materials and Methods section to the third section. In addition, the two subtitles, miRNA sequence and phylogenetic analysis, were used in this section.

Round 2

Reviewer 1 Report

The revised version of the paper is acceptable as it addressed concerns.

Author Response

Thank you for your comments.

Reviewer 2 Report

I see the great improvement in the revised manuscript.

All of your responses are reasonable.

Author Response

Thank you for the comment.

Reviewer 3 Report

The disorders called now ‘Anti-NMDA receptor Encephalitis’ or in short ‘NMDA Encephalitis’ , which predominantly affects children and young adults, occurs with or without tumour association, and responds to treatment, but can relapse.

The presence of a tumour (usually an ovarian teratoma) is dependent on age, sex and ethnicity, being more frequent in women older than 18 years, and slightly more predominant in black women than in white women.

Patients treated with tumour resection and first-line immunotherapy: Corticosteroids, IVIG, or plasma exchange, respond faster to treatment and less frequently need second-line immunotherapy: Cyclophosphamide or Rituximab, or both, than do patients without a tumour who receive similar initial immunotherapy.

The excellent, comprehensive and updated review on ‘Anti-NMDA receptor Encephalitis’, and on the anti-NMDA-R1 antibodies that cause this disease, have been published by Dalmau’s group and published in 2011 in ‘The Lancet’. For the benefit of the readers, and as a special tribute to Dalmau et al. (2007, 2008, 2011) for their pioneering and key discoveries over the years in regards to ‘Anti-NMDA receptor Encephalitis’ (Iizuka et al. 2008).

Therefore, Author should improve and correct many comments and parts in the sections of Introduction and discussion, based on the previous and recent medical consensus.  

Furthermore, if this manuscript is article, author should improve according to instruction.

Additionally, the references should be reduced.

Author Response

Thank you very much for the comments. I have revised the paper according to your comments. The point-by-point responses are provided as follows. All changes of this manuscript are highlighted with track changes in Word. In addition, the writing of this version has been edited by MDPI editing service.

Therefore, Author should improve and correct many comments and parts in the sections of Introduction and discussion, based on the previous and recent medical consensus.

Response: Thank you for the comment. I added the following sentences in the Introduction section and Discussion section.

Page 3, lines 2 and 3. “which was described and defined by Dalmau and colleagues [1]”.

Page 3, lines 9 and 10. ‘The disease is more prevalent in women and about 37% of patients are younger than 18 years [2]’.

Page 3, lines 17 and 18. ‘More updated discussion and review of this disease including animal models are provided by the experts of this field [2].’

Page 3, lines 20-22. ‘Most patients with anti-NMDAR encephalitis respond to immunotherapy and the immunotherapies used, timing of improvement, and long-term outcome haved been studied [11].’

Page 3, lines 2-6 ↑. ‘In addition, a grading score predicting neurologic function one year after diagnosis of anti-NMDAR encephalitis was constructed [14]. The functional status of 382 patients one year after diagnosis was studied, and factors associated with poor status were identified.’

Page 15, lines 3-5 ↑. ‘In addition, identifying more prognostic biomarkers and developing efficient treatments are the two most important issues in future studies [2].’

Furthermore, if this manuscript is article, author should improve according to instruction.

Response: I checked the instructions for Authors. The organization of the article should be “The Introduction, Materials and Methods, Results, Discussion, Conclusions, Supplementary Materials, Author Contributions, Funding, Acknowledgments, Conflicts of Interest”. In my first revision, although I added a supplementary file, I did not include the ‘Supplementary Materials’ statement in it.

I add this in this version. In addition, according to the instructions, the second section should be the Materials and Methods section. However, it needs more space to introduce the searched miRNA biomarkers. It may not be suitable to include this content in Introduction section and Materials and Methods section. Therefore, I added a section ‘MicroRNA’ to include these searched results before the Materials and Methods section.

Additionally, the references should be reduced.

Response: I have removed 10 references. But in this version, since I included some recent studies of anti-NMDA receptor encephalitis, I added two new references in this version.